# Global Trends in Halal Food Standards: A Review

**DOI:** 10.3390/foods12234200

**Published:** 2023-11-21

**Authors:** Junaid Akbar, Maria Gul, Muhammad Jahangir, Muhammad Adnan, Shah Saud, Shah Hassan, Taufiq Nawaz, Shah Fahad

**Affiliations:** 1Department of Islamic & Religious Studies, The University of Haripur, Haripur 22620, Pakistan; junaid.akbar@uoh.edu.pk; 2Department of Islamic Studies, Women University Mardan, Mardan 23200, Pakistan; mariagull317@gmail.com; 3Department of Food Science Technology, The University of Haripur, Haripur 22620, Pakistan; m.jahangir@uoh.edu.pk; 4College of Food, Agricultural and Environmental Sciences, The Ohio State University, Columbus, OH 43210, USA; adnan.25@osu.edu; 5College of Life Science, Linyi University, Linyi 276000, China; 6Department of Agricultural Extension Education and Communication, The University of Agriculture, Peshawar 25000, Pakistan; janslandscaper@gmail.com; 7Department of Biology/Microbiology, South Dakota State University, Brookings, SD 57006, USA; taufiq.nawaz@jacks.sdstate.edu; 8Department of Agronomy, Abdul Wali Khan University Mardan, Mardan 23200, Pakistan; 9Department of Natural Sciences, Lebanese American University, Byblos 1401, Lebanon

**Keywords:** halal standards, accreditation bodies, halal certification bodies, stunning, animal slaughtering

## Abstract

The demand for ethical foods is rising, with halal foods playing a significant role in this trend. However, halal standards vary globally, which can have substantial implications. Multiple Halal Certification Bodies (HCBs) can approve food products but they often prioritize national regulations over international alignment. To explore the similarities and differences in halal standards, we conducted a critical analysis of various standards, including Pakistan’s halal standards, the Standards and Metrology Institute for Islamic Countries, Majlis Ugama Islam Singapore, Majelis Ulama Indonesia, GCC Standardization Organization, Jabatan Kemajuan Islam Malaysia, ASEAN General Guideline, and the halal standards of Thailand, Iran, and Brunei, through a literature survey. While some commonalities exist, differences stemming from various Islamic schools of thought pose challenges for regulators, consumers, and food producers. Controversial issues include stunning, slaughtering, aquatic animals, insects, and labeling requirements. For example, all standards except the GSO allow non-Muslim slaughterers, and stunning is permitted in all standards except those of Pakistan. These disparities underscore the need for standardization and harmonization in the halal food industry to meet the growing demand for ethical foods.

## 1. Introduction

According to a January 2023 report, Muslims make up 02 billion, i.e., 25.0% of the world’s total population [1]. “Halal” is one of the basic requirements for Muslims, especially concerning food, pharmaceuticals, and cosmetics. The demand for halal products is ever-increasing with the increased awareness of consumers. A recent survey representing 38,000 Muslims concluded that 96% of Muslims, while traveling abroad, want halal food [2]. As halal is a mandatory dietary law in Islam, consumer trust has recently shifted towards third-party conformity (Halal Certification) of food products manufactured globally (in Muslim-majority or non-Muslim majority countries). For this purpose, there are many halal certification bodies worldwide. Countries with Muslim majority populations (Islamic Countries) have developed halal standards per their indigenous requirements. Among these critical players in the halal market are the Standards and Metrological Institute of Islamic Countries (SMIIC), Majlis Ulama Islam Singapura (MUIS), Majlis Ulama Indonesia (MUI), Gulf Standardization Organization (GSO/GCC), Jabatan Kemajuan Islam Malaysia (JAKIM) and Pakistan Standards and Quality Control Authority (PSQCA).

## 2. Materials and Methods

As there is no consensus on a single halal standard nor one accreditation council, Halal Certification Bodies (HCBs) have adopted single or multiple halal standards and have often been accredited by various halal accreditation bodies. Detailed evaluations have found that 85 HCBs from 47 different countries have been accredited by JAKIM, 53 HCBs from 39 countries by MUIS, 45 HCBs from 26 countries were accredited by MUI, 32 HCBs from 21 countries have been accredited by SMIIC [3], and 19 HCBs have been approved by GCC [4]. The lowest number of HCBs accredited currently is 8, by the Pakistan National Accreditation Council (PNAC), operating from within Pakistan only [5]. In addition, Brunei, Thailand, and Iran have also developed halal standards, whose details will be discussed in this study.

This study aims to conduct a comparative study of global halal standards and critically evaluate them from a consumer perspective. To achieve this goal, library research methodology has been adopted, and literature on halal standards and the halal standards of different countries have been reviewed (such as JAKIM, SMIIC, MUIS, MUI, GSO, Pakistan, Brunei, Iran, and Thailand). All of the information regarding the standards was retrieved from those published and available on the websites of certification bodies concerned. The citations for the same are given for reference, in addition to the research articles for comparison.

## 3. Global Halal Standards

### 3.1. Singapore (MUIS) Halal Standards

Singapore’s Islamic religious council, Majlis Ugama Islam Singapore (MUIS), has developed halal standards with the help of SPRING Singapore (Standards, Productivity, and Innovation Board). The halal certification in Singapore is overseen by the Majlis Ugama Islam Singapore Standards Committee, which includes religious scholars, industrialists, and government employees. The committee has multiple objectives, such as providing business and trade opportunities, ensuring consistency in compliance with the certification conditions, and offering technical and religious guidance for halal certification [6]. There are two standards regarding halal; the first, MUIS-HC-S001 (approved on 21 May 2005), is a guideline for preparing and providing halal food (Islamic Religious Council of Singapore, 2005). The second is MUIS-HC-S002 (approved by the Technical Committee of MUIS on 13 May 2005), a guideline for compliance and development of the halal quality management system [7].

### 3.2. OIC/SMIIC

The Standards and Metrology Institute for Islamic Countries (SMIIC) is a governmental organization that sets standards in collaboration with the Organization of Islamic Countries (OIC) to ensure consumer safety and product quality for strengthening the market position of OIC member countries and promoting free trade. The preparation of the SMIIC halal standards began in 2008, and the first edition of these standards was completed in 2011 by SMIIC’s Standardization Expert Group (SEG), comprising 39 member countries of OIC and the International Islamic Jurisprudence Academy (IIFA).

SMIIC’s halal food standards are known as OIC/SMIIC 1: 2019 (the second edition was published on 31 July 2019), which define the basic requirements to be implemented at any stage of the food chain. This standard provides general information to the parties concerned, such as consumers, product manufacturers, and diagnostic agencies, to guarantee that halal food products are manufactured according to Islamic rules. The general requirements of the standard consist of several different aspects regarding the processing of halal products. These include principles of slaughter and food products throughout the supply chain, including services, health, and food protection, as well as validation and verification, identification, traceability, and legal requirements, among other aspects [8].

Published around the same time, the OIC/SMIIC 2: 2019 (on 22 July 2019) is a set of rules and regulations developed to satisfy halal certification bodies and is required to implement halal certification activities [9]. Additionally, OIC/SMIIC 3: 2019 (published on 22 July 2019) set out the general guidelines and procedures for reviewing and approving halal certification bodies at the regional and international level for halal accreditation bodies [10]. Furthermore, OIC/SMIIC 6: 2019 (published on 1 August 2019) has unique requirements for places where halal food and beverages are prepared, stored, or served [11].

In sequence and a step ahead of these, OIC/SMIIC 17-1: 2020 (published on 4 November 2022) describes the requirements of a supply chain management system that ensures the integrity of halal products/goods through various modes of transportation [12]. Additionally, OIC/SMIIC 17-2: 2020 (4 November 2020) defines the requirements of supply chain management systems that ensure the halal integrity of the goods/products in the warehouse [13]. The OIC/SMIIC 17-3: 2020 (on 4 November 2022) was introduced to define the supply chain management system requirements that ensure halal integrity at the point of sale of goods/products [14].

The OIC/SMIIC 18:2021 (10 July 2021) is a requirement for a halal quality management systems [15]. In addition, OIC/SMIIC 22:2021 (published on 17 June 2021) contains requirements and methods for testing edible gelatin [16].

The standard document number OIC/SMIIC 24: 2020 (published on 23 June 2020) guides manufacturers in using various chemicals to prepare halal food. To accomplish this purpose, a list is provided in the standard, which clarifies whether these ingredients are halal, doubtful, or haraam. There are also requirements concerning the labeling of products, including a requirement to reference the products’ ingredients [17].

In addition, the OIC/SMIIC 33: 2020 (published on 7 June 2022) is among a series of halal conformity assessment standard documents that deal with the basic principles of halal certification and provide guidelines for understanding, developing, operating, and maintaining certification schemes [18].

The OIC/SMIIC 34: 2020 (published on 7 June 2020) includes the principles and general information for halal certification bodies to certify individuals who are part of halal-related activities and maintain and develop a certification scheme for such individuals [19]. It is important to note that this document was the first official document utilizing scheme-based certification guidelines. Another contribution of this manual is to address the growth in laboratory assurance of various halal products.

The OIC/SMIIC 35: 2020 (published on 7 June 2020) specifies the general requirements for halal testing laboratories. All organizations that carry out laboratory activities fall within the scope of this standard [20]. Additionally, the document OIC/SMIIC 36: 2020 (published on 7 June 2020) specifies standard requirements for the competence of companies dealing with halal scalability checking-out schemes and for the development and operation of halal talent-testing schemes. These requirements are famous for all sorts of halal proficiency-testing schemes [21].

### 3.3. Halal Standards of Malaysia (JAKIM)

The Malaysian Standard Development System drafted the Malaysian Standards with the aid of several national organizations such as The National Board of Standardization and Quality (SIRIM), the Federation of Manufacturers Malaysia, University Putra Malaysia (UPM), The International Islamic University (IIUM), University Technology Mara (UiTM), The Agricultural Research and Development Institute of Malaysia, The Malaysia Quality Institute, The Department of Veterinary Services, The Department of Standards Malaysia, The Department of Science and Technology, and The Ministry of Health and Science. The Malaysian Department of Islamic Development, Jabatan Kemajuan Islam Malaysia (JAKIM), was established in 1982. JAKIM was given the responsibility by the Prime Minister to oversee Islamic affairs and create halal-awareness programs for food producers, distributors, and importers. In addition to this, JAKIM also supervised food factories and hotels. Since then, JAKIM has been responsible for enforcing halal laws [22].

Malaysia’s halal standards regarding halal food are documented in MS 1500:2009 (published in 2018), and are related to the production, preparation, storage, and supply of halal food [23]. Similarly, three parts of MS 2400 were published in 2010. The first part, MS 2400-1: 2010, concerns the principles of the management system related to the transportation of goods, and MS 2400-2: 2010: the principles of the halal warehousing management system. The MS 2400-3: 2010: concerns the principles of the management system related to the sale of halal products [24].

There was an addition to these regulations in MS 2200-2: 2013 (developed in 2013), which related to using animal skins, bones, and hair in Muslim consumer products [25]. The MS 2565: 2014 (published in 2014) concerns the packaging instructions for the halal production system [26].

### 3.4. Halal Standards of Indonesia (MUI)

Majelis Ulama Indonesia (MUI) founded LPPOM-MUI on 6 January 1998 to oversee halal-related matters for food, medicine, cosmetics, and other products. LPPOM-MUI has been instrumental in effectively formulating halal food, beverages, cosmetics, and medicines. It serves as a national and international halal certification body, ensuring that all certified products meet halal standards. LPPOM has agreements with the following entities: National Agency for Drugs and Food Control, Bogar University of Agriculture, Ministry of Religious Affairs, Ministry of Agriculture, Ministry of Cooperative, and Small Scale Industry [27].

LPPOM-MUI standards regarding halal are HAS-23000:1 requirements for halal certification. HAS-23000: 2 deals with requirements of halal certification for meat-processing factories. HAS 23000-3 describes halal certification requirements for hotels and food services. HAS 23103 contains instructions for methods of halal assurance systems in slaughterhouses. HAS 23201 defines the requirements for halal food items [28].

### 3.5. Halal Standards of the GCC Standardization Organization (GSO)

The GCC (Gulf Cooperation Council) Standardization Organization (GSO) is a regional standardization organization established through a resolution of the GCC Supreme Council. The meeting occurred on 30–31 December 2021 and began work in May 2004. The governments of the United Arab Emirates, Kuwait, Bahrain, Saudi Arabia, Oman, Qatar, and the Republic of Yemen joined the organization in January 2010. The goal of this organization was to combine quality activities and establish cooperation with the standard-making bodies of member countries [29].

The halal standards of the GSO are detailed in GSO 2055-1:2015 (approved on 5 November 2015), which specifies the general requirements for halal food that will be applicable at any stage of the halal food chain, including preparation, packaging, labeling, receipt, transportation, distribution, storage, display, handling, and halal food services [30]. Additionally, the GSO 993:2015 standard document is related to slaughtering animals (including birds) as per Islamic law [31].

GSO 2055-2:2021 (approved on 1 July 2021) is a general requirement for halal certification bodies [32]. GSO 2055-3:2021 (approved on 1 July 2021) is a general requirement for halal accreditation bodies certifying halal certification bodies [33].

GSO 2468:2021 (approved on 1 July 2021) is a halal food management system requirement for cargo chain services [34].

GSO 2652:2021 (approved on 1 July 2021) describes general guidelines for preparing and handling halal packaging. It works as an essential requirement for the halal packaging of halal products [35].

GSO 2470:2021 (approved on 1 July 2021) applies to data that is required at the management system level to ensure the protection of halal identity in the retailing phase of halal products [36].

GSO 2469:2021 (approved on 1 July 2021) relates to the requirements for halal integrity in the management system during the entire process from the receipt to the delivery of goods and cargo warehousing and related activities [37].

GSO 2670:2021 (approved on 1 July 2021) provides practical guidelines for industries related to the use of animal bones, skin, and hair as per Islamic law [38].

### 3.6. Pakistani Halal Standards

The halal system in Pakistan is currently under multiple stake-holding entities supervised by The Ministry of Science and Technology. Different organizations, including The Pakistan Standard and Quality Control Authority (PSQCA), Pakistan National Accreditation Council (PNAC), Pakistan Halal Authority (PHA), provincial food regulatory bodies, and Halal Certification Bodies (HCBs) are functionally engaged with the Halal system in various capacities. At the regional level, four food authorities operate: Punjab Food Authority, Sindh Food Authority, Khyber Pakhtunkhwa Food Safety and Halal Food Authority, and Balochistan Food Authority.

Pakistan Standards and Quality Control Authority covers efforts to develop halal standards. Various technical committees have been set up to develop these standards. These committees comprise representatives from the organizations above and personnel from research and academia.

The government of Pakistan established the Pakistan Standards and Quality Control Authority (PSQCA) in 1996. The organization was assigned the task of formulating and developing standards (including but not limited to halal) for various goods and services, ultimately leading to the advancement of the national economy, improving the health and safety of the people, advancing standards internationally for the benefit of consumers, and in domestic and international trade to provide facilities [39].

The halal standards of the Pakistan Standards and Quality Control Authority (PSQCA), known as PS: 3733-1: 2019, are related to the basic requirements of the Halal Management System and its terminologies. This standard has four parts, approved on 27 December 2018. Among these, the PS: 3733-2: 2019 is the requirement of halal management systems for any organization in the food chain, the PS: 3733-3: 2019 are the requirements of halal management systems for slaughtering animals, the PS: 3733-4: 2019 are the requirements of halal management systems related to chickens/birds [40] and PS: 247: 2013 is about edible halal gelatin [41].

### 3.7. Halal Standards of Brunei Darussalam

Various government organizations work together in Brunei Darussalam to address food issues related to halal food. As the import and export of food are under the jurisdiction of the Royal Brunei Custom Excise Department, which belongs to the Ministry of Finance, the Halal Food Control Division for halal food certification is under the Ministry of Religious Affairs. Halal certificates for slaughterhouses and restaurants and halal labels for various halal products are provided by the Brunei Religious Council [42]. In addition, halal food standards were developed by a technical committee set up by the authority of the religious council of Brunei.

The halal standard of Brunei Darussalam is called PBD 24: 2007. This standard relates to the production, manufacture, supply, distribution, and storage of halal food. This standard provides guidelines for preparing and handling halal food and serves as a basic requirement for food products and businesses in Brunei [43]. In addition to this standard, four guidelines are used in Brunei. BCG Halal—1 (developed in 2007) is a guideline for halal certification [43], the BCG Halal—2 are the guidelines for compliance and auditing [44], BCG Halal—3 are the guidelines for the auditing of halal surveillance, and finally, the BCG Halal—4 are halal brand guidelines for marketing products in the global market [45].

### 3.8. Iranian Halal Standard

Iran’s halal standards are set by the Institute of Standards and Industrial Research of Iran (ISIRI). The organization was founded in 1960 and became a member of the International Organization for Standardization (ISO) in the same year. It has been under the direct supervision of the President of Iran since 2011. Its primary purpose is to develop, publish, and identify national standards [46]. Iran’s halal standard is called ISIRI 12000 (published in 1992), and is a guideline for all halal food supply chain stages which includes eleven articles about halal [47].

### 3.9. Halal Standards of Thailand

The Central Islamic Committee of Thailand (CICOT) is a non-profit organization established under the Administration of Islamic Organization Act B.E.2540, A.D 1997. In 2013, CICOT established the Halal Standards Institute of Thailand to develop standards for approving halal products. At the national level, the organization has the following responsibilities: to implement the Thai Halal Products Standard and ensure that the standard of the product is developed under Islamic principles and international standards, to implement the halal logo for halal products, to act as a halal accreditation body for the approval of halal certification bodies, and to coordinate and oversee halal affairs and related units to make halal product standards effective [48].

The Halal Standard Institute of Thailand has established various standards for halal products. These include THS 1435-1-2014, which specifies the requirements for verifying the halal production process of a product. THS 1435-2-2014 outlines the certification requirements for halal slaughterhouses and slaughter. THS 1435-3-2014 provides guidelines for halal certification, while THS 1435-4-2557 provides instructions for applying for halal marks and using them on products and packaging. Lastly, THS 1435-5-2014 defines the rules for conducting product audits and fee inspections for holding certificates [49]. General guidelines for halal products are outlined in TAS 8400-2007. This standard covers the preparation, processing, packaging, storage, presentation, distribution, and labeling of halal food, as well as food safety [50].

## 4. Similarities and Differences in Global Halal Standards

The halal standards from every accreditation body mainly consist of Shari’ah (religious obligations) and administrative clauses. The Shari’ah section generally mentions the definitions and rules of Shari’ah issues such as halal, haraam, hajis, and slaughter, among others. The differences in the standards prevailing in different countries are due to the affiliations of masses with varying schools of Islamic thought. The issues relevant to the jurists include explaining the Qur’an and Sunnah and differences in jurisprudence opinions.

On the other hand, the administrative clauses deal with halal assurance management-related procedures, management systems, validation, and verification through facts. In the case of administrative clauses, there is also a difference as the standards need to be tailored to the country’s needs, and at some level, these have to refer to international requirements as well. The main differences between these standards are discussed below.

### 4.1. Halal Slaughter

Slaughtering is a common but most critical concern from a halal certification point of view. However, in general, it seems there is a consensus that for halal slaughtering, the animal is categorized as halal only if the method used is an Islamic way of slaughtering, and only if the slaughterer is a Muslim or (if not available) may be a person of the book (practicing Christian, Jew) or Zoroastrian. Still, there are huge differences among various Islamic schools of thought about this issue. Hence, multiple regulatory and accreditation bodies have developed their standards for slaughtering. Here, the focus is on the slaughterer primarily, where according to OIC/SMIIC, Indonesia [51], Malaysia [52], Singapore [7], Thailand [50], Iran [47], and Pakistan [53], the slaughterer must be practicing Muslim and familiar with Islamic laws regarding slaughtering. Interestingly, according to Section 4.2.1 of GSO 993, being a Muslim is not a condition for slaughter; a slaughterer can be a Christian, a Jew, or a Muslim [54] (See Table 1).

There are also age restrictions in Malaysian and Indonesian standards; the slaughterer must be at least 18 years old, intelligent, with a healthy medical record, and the number of slaughterers daily must be commensurate with the number of animals slaughtered [51].

### 4.2. Stunning

The stunning process consists of an electric, chemical, or mechanical shock to animals, rendering them immobile or unconscious, with or without killing them before slaughter. There are generally three methods of stunning. Electrical stunning, captive stunning, and the use of carbon-dioxide gas. Due to stunning, the animal’s blood is drained until it becomes unconscious and dies from the lack of blood [55].

#### 4.2.1. Halal Standards Regarding Stunning

The third part (Section 4.1.6) [56] and the fourth part (Section 4.6) of the Pakistan Halal Standard clearly state that any stunning before slaughtering is forbidden, and the rules also apply to meat imported from other countries [53]. According to Iran [47], Brunei [57], and Singapore [7] standards, stunning is permitted if it does not cause death, decapitation, or permanent damage to the animal. However, no further conditions or details are specified for stunning. Similarly, stunning is allowed in Indonesia’s halal standard and in SMIIC 1: 2019 [8], where only electrical stunning is allowed on severe animal resistance, but still mechanical stunning is prohibited (See Table 2).

In SMIIC 1: 2019, the stunning of small animals is not mentioned, while the standard time and amount of current are specified for stunning bigger animals [58].

The ASEAN Guideline allows two specific stunning methods, i.e., mechanical and electrical. The conditions for stunning are that stunning equipment will always be at the disposal of a Muslim who is a supervisor/trained Muslim slaughterer or from the halal certification authority, and the stunning should be temporary and in such a way that the animal can return to its original state within five minutes.

#### 4.2.2. Mechanical Stunning

Mechanical stunning refers to firing a bolt (may be penetrative or non-penetrative) through the skull of the animal using a pneumatic device or a pistol [59]. Mechanical stunning should be used only for cows and buffalos. After slaughtering the animal and removing skin from its skull, it should be examined. If a permanent wound, scratched or broken, is found on its skull, this animal will be separated from other animals as it will not be considered halal.

#### 4.2.3. Electrical Stunning

Electrical stunning passes a current of electricity through the brain of the animal. The certification authority determine the amount of current used. An electrical stunner acts as ‘slaughter in-charge’ with the permission of the government/authority. Only ‘water stunning’ is used for small animals (poultry). For the slaughter of halal animals, stunning is allowed only on part of the head [60]. The stunning methods and conditions in the halal standard of Malaysia [61] and Thailand [50] are similar to the ASEAN Guidelines.

The amount and duration of current used on animals may vary according to their weight, but a trained Muslim or responsible authority will determine this. The GSO 933 disapproves of all forms of stunning and anesthesia for animals; if necessary, larger animals may only be given a low voltage shock on their heads. If they are electrocuted, they are considered dead. In comparison, all types of fainting and electric shocks are prohibited for the handling of birds. Captive bolt pistols, hammers, or air blowing are not permitted to anesthetize animals [54] (See Table 3).

### 4.3. Mechanical Slaughter

Mechanical slaughtering is dependent on animal type and size, such as for larger animals, e.g., cows, buffalos, goats, and sheep. These are first stunned and then brought by the machine to a place where a man can cut its throat with a sharp knife; then, the animal is hung and the machine removes its skin for further cleaning, butchering, and packaging. The second case is for small animals (poultry). The chickens and birds are brought to the slaughterhouses and then hung by their feet in a machine. Before slaughtering, they are stunned to reduce their resistance. After the stunning process, they are brought to a sharp blade, and with this blade they are slaughtered. With this machine, the throat is cut in such a way that the neck is not entirely separated [62].

#### Halal Standards Regarding Mechanical Slaughter

Mechanical slaughtering of both bigger and smaller animals is prohibited by Pakistan [53] and Brunei standards [43].

The ASEAN Halal Guideline allows the mechanical slaughter of chickens according to certain conditions: the operator of the machine will be a Muslim who will recite the word “Bismillah” (i.e., with the name of Allah) before operating the machine. A Muslim slaughterer who operates the machine will not leave the place of slaughter; if, for some reason, he needs to leave, another Muslim would take his place, recite “Bismillah”, and start the machine again. The slaughterer will look at each bird separately, and if a bird has been left out of the mechanical slaughter or has not been slaughtered properly, he will slaughter it by hand [60].

The Iran Halal Standard allows the mechanical slaughter of chickens [47]. Yet interestingly, the Thai Halal Standard generally does not recommend the mechanical slaughter of poultry, but where mechanical slaughter is required, it is permitted [50].

The conditions of mechanical slaughter in Iran and Thailand are like those in the ASEAN guidelines. The mechanical slaughter of large animals is not mentioned in these three standards.

The conditions of mechanical slaughter in OIC/SMIIC [58] and GSO are like the above standards, but specific animals are not specified [54]. Likewise, Singapore’s halal standards do not specify the conditions and procedure for mechanical slaughter, except that the machine operator recites “Bismillah” before running the slaughtering machine [7] (See Table 4).

### 4.4. Seafood

According to the OIC/SMIIC [58], GSO [63], Singapore [7], United Arab Emirates [64], ASEAN Guideline [60], and Thailand’s [50] halal standards, all aquatic animals are edible except for poisonous or harmful ones. Such aquatic habitats are not permitted unless their toxicity is eliminated. Similarly, the Brunei Halal Standard declares all aquatic animals to be halal, whether caught or found dead, except those harmful to human health. But the animals that live both on land and in water, such as frogs, crocodiles, and crabs, are not allowed [43]. In the same way, according to the Malaysian Halal Standard, all aquatic animals are halal except those that are poisonous and harmful. Those aquatic animals are also haraam, as they live or are raised in filth and consume impure food [65]. Like others, Iranian halal standards forbid all aquatic animals except fish with scales and shrimps are permissible [47].

According to the Pakistan Halal Standard, all aquatic animals are halal, except under the conditions that the death of the animal is not associated with fishing, but rather with causes like disease, competition, cannibalism, old age, predation, pollution, or any other natural factors that cause death, and that the animal is not poisonous or harmful to human health. In Pakistan halal standards, all aquatic animals except fish are makrooh-e-tahreemi (which has been declared undesirable by Shari’ah and is very close to Haraam) [40] (See Table 5).

### 4.5. Use of Insects

In Pakistan [40] and GSO 2055-1 [63], all insects are haraam except locusts and the parts of the honey bee that fall into honey. The ASEAN Guideline [60], and Thailand’s [50] and Malaysia’s [7] halal standards prohibit ugly insects like lice and flies.

In OIC/SMIIC [58] and Brunei’s standards, only locusts are permitted [57]. Similarly, as per Singapore’s halal standards, all insects are forbidden except locusts, crabs (not poisonous), and dabb lizards/Uromastyx (spiny-tail lizards) [7], whereas all insects are prohibited in the halal standards of Iran [47] (See Table 6).

### 4.6. Standards about Filthiness (Najis)

In Pakistan’s [40] and ASEAN [60] halal standards, najis is defined as something hateful or mixed with substances such as alcohol, blood, pus, ova, sperm, etc. There are three types of najis in the halal standards of Malaysia: (1) Mughallazah (considered as severe najis), (2) Mukhaffafah (considered as light najis), and (3) Mutawassitah (considered as medium najis).

#### 4.6.1. Mughallazah

Dogs, pigs, and any substance that is extracted from their bodies.

#### 4.6.2. Mukhaffafah

Less impure than the urine of a child two years old or younger who solely consumes breast milk.

#### 4.6.3. Mutawassitah

Medium types of impurity such as vomit, pus, blood, Khamar, and milk from humans and other animals are not considered najis by Malaysian standards. The Malaysian standards only describe how to clean the najis classed mughallazah. To remove najis, the item should be washed seven times, including once in muddy water. The soil should be soluble in the water.. It should be washed seven times so that the water it was washed with the first time is well-drained, and then washed a second time [65].

The types of najis are also mentioned in the halal standards of Thailand, and the method of cleaning halal material is also mentioned. To purify the mukhaffafah najis, one should remove the najis item and sprinkle water on it. For the average najis, the item should be removed and washed three times with water, thereby eliminating the smell, taste, and color of the impurities. To clean mughallazah najis, it should be washed seven times. The first of these seven washes should be performed with groundwater mixed with soil [50].

There are three types of najis; as per the Singapore halal standard, the definitions of najis mughallazah and mutawassitah and their respective purification methods are similar to those in Thai standards. While najis mukhaffafah are slightly stained on surfaces on which stain removal is difficult, a small amount of blood may remain after washing the meat or the stomach contents of a small halal animal. It does not need to be cleaned or removed if it is difficult to do so [7].

There are two types of najis mentioned in Iran’s halal standards.

Najis-ul-Ain, which cannot be cleansed like dogs, pigs, blood, and other materials.Nijast, which is not najis in itself but has become najis, like a blood-stained vessel [47].

In OIC/SMIIC, the definition of najis is not stated while two types of najis, visible and invisible, are mentioned, and the purification method is written in the standard [58] (See Table 7).

### 4.7. Labeling/Packaging

Labeling is not mentioned in the ASEAN General Guideline [60]. At the same time, halal labeling and packing standards have been suggested in Iran [47], Brunei [57], UAE [64], Thailand [50], Singapore [7], Malaysia [65], and Pakistan [53].

Iran’s standards for halal-quality packaging material is that it should not be harmful to the health and safety of the consumer. The packaging material should be non-solvent and not contaminated with impure elements. Meanwhile, Malaysia, UAE, and Brunei have similar standards regarding labeling. These include labeling/packaging materials not made from materials that are harmful to human health or not halal. Labeling/packaging material should be kept separate from non-halal materials at every stage (preparation, storage, transportation), and special care should be taken for the sake of hygiene. The packaging/labeling should include the product name, weight, manufacturer’s name, address, and importer’s name and address on the imported food. The UAE and Malaysian standards also set out additional requirements for meat packaging. These include meat type, date of slaughter, and date of processing.

The characteristics of the packaging material in Thai halal standards are similar to the above standards. The noteworthy aspect here is that the halal mark on any product must a halal certification body. If the product contains any forbidden ingredient, animal, or fish, it must be specified on the labeling. Meanwhile, in OIC/SMIIC [58] and Pakistan standard PS3733 [40], detailed terms for packing and labeling are mentioned. Exceedingly, the following conditions are also mentioned:The packing process should be carried out at appropriate temperatures so as not to affect the product’s quality.The product’s name should not be mixed with the haraam product’s name.If the product is being used without packaging, writing the instructions at the point of sale is essential.The place and method of slaughter, import/export, or distributor address should be present on the meat.The method of product use and whether Genetically Modified Organisms (GMO) were used must be written on the label. Also, the type of product (fresh, dried, or smoked) and the sources of the product are necessary to report.A veterinary certificate certifying the health of the animal must be included. The ink used for stamping the meat should not be harmful to health. If the meat is imported from another country, a certified slaughterhouse certificate is required to prove that the animal’s health has been checked before and after slaughter and that it is not harmful to human health. In addition, the certificate should include the average age of the animal, its weight and type, the date of slaughter, the name of the country, stunning method and hand slaughter method, and all details related to the slaughter must be included [53].

## 5. Challenges Faced by Halal Certification Bodies and Their Solution

Halal certifiers face challenges in implementing varying halal standards. Here are some common issues and potential solutions that can help address them:

Inconsistencies and confusion can arise due to the absence of a universally recognized halal certification standard. To promote consistency and facilitate international trade, international halal standards should be established and adopted, such as those established by the Organization of Islamic Cooperation (OIC).

There is often a lack of clarity around what is considered halal in Muslim communities and among scholars. To address this, certification bodies can collaborate with religious organizations and scholars to establish a consensus on halal standards. Clearly defined criteria and guidelines can help reduce confusion and ensure consistency.

It can be difficult to ensure that products comply with halal standards because they are sourced from different countries. However, developing blockchain and traceability technologies can provide transparency in the supply chain. To achieve this, businesses should maintain detailed records and documentation of their supply chain.

The lack of consumer awareness about halal certification could be addressed through public awareness campaigns leveraging digital platforms, social media, and labeling [66].

## 6. Conclusions

Consumers’ interest and increasing demands for halal products encourage manufacturers to comply with halal standards. Different countries have set their own halal standards, such as Pakistan and Iran. In contrast, some standards are acceptable in broader regions and may be leading some authorities over others, such as the JAKIM, MUI, and GSO. Unfortunately, no single halal standard is recognized worldwide, so there is enormous consumer and producer concern about what standards to follow and the demands of different countries. These standards generally have some differences regarding slaughter, stunning, mechanical slaughter, types of impurity, and the halal status of marine animals and insects.

However, the reason for their differences is often the affiliation of jurists with different Islamic sects. According to GSO, the slaughterer can be Christian or Jewish, while other standards require the slaughterer to be a Muslim. Similarly, the principles of standards differ from each other. Some contain discussions of the slaughterer’s age and of stunning and mechanical slaughter. Pakistan’s standards allow only fish to be eaten, while different standards permit the consumption of various aquatic animals. Only locusts are allowed in Pakistan’s, Brunei’s, GSO, and OIC/SMIIC standards, while other standards may also use insects other than locusts. However, importantly, in all of these standards, najis has been mentioned, though there are slight differences in the methods of cleaning and its types. Packaging and labeling principles are included in all standards, except in Asia’s general guidelines. The mutual recognition of halal standards by accreditation bodies for import/export and permission to function in the counterparts of respective regions is a workable solution.

## Figures and Tables

**Table 1 foods-12-04200-t001:** Differences and similarities in standards concerning slaughterer.

Slaughterer	Pakistan	SMIIC	GSO	Singapore	Indonesia	ASEAN	Malaysia	Thailand	Iran
The slaughter must be a Muslim	√	√	×	√	√	√	√	√	√
The slaughterer may be from the Semitic religion.	×	×	√	×	×	×	×	×	×
The slaughterer must be 18 years old.	-	-	-	-	√	-	√	-	-

(√: allowed, ×: not allowed, -: no specific item is mentioned).

**Table 2 foods-12-04200-t002:** Differences and similarities in standards concerning stunning.

Stunning	Pakistan	SMIIC	GSO	Singapore	Indonesia	ASEAN	Malaysia	Thailand	Iran	Brunei
Stunning is allowed	×	√	√	√	√	√	√	√	√	√
Stunning of large animals	×	√	√	-	-	-	√	√	-	-
Stunning of small animals	×	-	×	-	-	-	√	√	-	-
Duration/volt of current	×	√	-	-	-	-	√	√	-	-

(√: allowed, ×: not allowed, -: no specific item is mentioned).

**Table 3 foods-12-04200-t003:** Comparison of the ampere and duration of current for animals in SMIIC, Malaysia/Thailand/ASEAN, and GSO standards.

Type of Animal	Current (Ampere)			Duration (Second)		
	SMIIC	Malaysia/Thailand/ASEAN	GSO	SMIIC	Malaysia/Thailand/ASEAN	GSO
Lamb	0.50–0.90	0.50–0.90	-	2.00–3.00	2.00–3.00	-
Goat	0.70–1.00	0.70–1.00	0.70–1.00	2.00–3.00	2.00–3.00	2.00–3.00
Sheep	0.70–1.20	0.50–0.90	-	2.00–3.00	2.00–3.00	-
Calf	0.50–1.50	0.50–1.50	-	3.00	3.00	-
Steer	1.50–2.50	1.50–2.50	-	2.00–3.00	2.00–3.00	-
Cow	2.00–3.00	2.00–3.00	2.00–3.00	2.50–3.50	2.50–3.50	2.500–3.50
Bull	2.50–3.50	2.50–3.50	2.50–3.50	3.00–4.00	3.00–4.00	3.00–4.00
Buffalo	2.50–3.50	2.50–3.50	2.50–3.50	3.00–4.00	3.00–4.00	3.00–4.00

**Table 4 foods-12-04200-t004:** Differences and similarities in standards concerning mechanical slaughter.

Mechanical Slaughter	Pakistan	Brunei	GSO	ASEAN	Singapore	Malaysia	Thailand	OIC/SMIIC	Iran
Permission of mechanical slaughter	×	×	√	√	√	-	√	√	√
Mechanical slaughter of small animals	×	×	-	√	-	-	√	-	√
Mechanical slaughter of large animals	×	×	-	-	-	-	-	-	-

(√: allowed, ×: not allowed, -: no specific item is mentioned).

**Table 5 foods-12-04200-t005:** Use of aquatic animals in different standards.

Aquatic Animals	Pakistan	Brunei	GSO	ASEAN Guideline	Singapore	Malaysia	Thailand	OIC/ SMIIC	Iran
All aquatic animals including fish are halal	×	√	√	√	√	√	√	√	Only fish with scales and shrimps
Other aquatic animals besides fish are halal	makrooh-e-tahreemi	√	√	√	√	√	√	√	×
Aquatic animals found dead are halal.	×	√	-	-	-	-	-	-	×
Those who live in filth and are fed najis are halal.	×	-	-	-	×	×	-	-	×

(√: allowed, ×: not allowed, -: no specific item is mentioned).

**Table 6 foods-12-04200-t006:** Uses of insects in different standards.

Insects	Pakistan	GSO	ASEAN	Thailand	Malaysia	Brunei	Singapore	OIC/SMIIC	Iran
Locust	√	√	√	√	√	√	√	√	√
Crab (non-toxic)	×	×	√	√	√	×	√	×	×
Dabb Lizard (spiny-tailed)	×	×	√	√	√	×	√	×	×
Non-Ugly insects	×	×	√	√	√	×	×	×	×
Ugly insects	×	×	×	×	×	×	×	×	×

(√: allowed, ×: not allowed).

**Table 7 foods-12-04200-t007:** Differences and similarities in standards concerning Najis.

Najis	Pakistan	Malaysia	Thailand	Singapore	Iran	ASEAN Guideline	OIC/SMIIC
Definition of najis	√	√	√	√	√	√	-
Three types of najis	×	√	√	√	×	×	-
Two types of najis	×	×	×	×	√	×	√
Method of cleaning najis	-	Only Najis mughallazah	√	-	-	-	√

(√: allowed, ×: not allowed, -: no specific item is mentioned).

## Data Availability

No new data were created or analyzed in this study. Data sharing is not applicable to this article.

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
