# Peer review of "Global Trends in Halal Food Standards: A Review"

_foods, 2023, doi:10.3390/foods12234200_

Round 1
Reviewer 1 Report
Comments and Suggestions for Authors
Please see comments in the attached file of the manuscript.

Comments on the Quality of English Language
minor English revision is required.
Author Response
|
|
Comments |
Compliance |
Line # |
|
1 |
Change in Title : Global Trends in Halal Food Standards: a review |
The title is Changed as suggested to: Global Trends in Halal Food Standards: A Review |
2 |
|
2 |
is this right? Email for a pakistan institute |
Email Updated |
9 |
|
3 |
stunning and .... |
Done |
30-31 |
|
4 |
As we are close to the end of 2023, this should be modified. |
Modified |
39-40 |
|
5 |
mushrooming .. |
The sentence changed with the correct word |
46 |
|
6 |
Please mention in brief the source of information. |
The source of the information is mentioned. |
66-71 |
|
7 |
Please mention in brief the mechanism by which mechanical stunning is |
Done |
349 |
|
8 |
please make sure it is gramatically correct |
Correction done |
363 |
|
9 |
if possible needs to be more clarified. |
Clarification done |
416-418 |
|
10 |
what does it mean? “Makrooh-e-Tahreemi” |
Clarification done |
420 |
|
11 |
Please provide a breif explanation for each “Terminoligies” |
Done |
437-439 |
Reviewer 2 Report
Comments and Suggestions for Authors
In the submitted manuscript, the authors reviewed the global trends in Halal food standards. The paper provides a critical analysis of various Halal standards, highlighting the similarities and differences between them. The study reveals that while some commonalities exist, differences stemming from various Islamic schools of thought pose challenges for regulators, consumers, and food producers. Overall, the text provides an overview of the Muslim population and their demand for Halal products. However, there are a few areas where the text could be improved:
Section 1:
- The text states that Muslims are expected to increase to 2.2 billion by 2023, but it does not provide a source for this claim.
- The text mentions a recent survey representing 38,000 Muslims, but it does not provide information on when the survey was conducted or who conducted it.
- The text lists several organizations involved in the Halal market, but it does not provide any information on what these organizations do or how they are involved in the market.
- please indicate the similarities and differences in Halal food standards across different countries.
- The authors should provide a more in-depth analysis of the challenges faced by halal certification bodies in implementing halal standards, as well as potential solutions to these challenges.
- The authors should state the impact of non-standardization on the economic growth of the halal food industry and international trade.
- The authors should address the implications of labeling requirements in Halal food standards.
- Provide more information on the organizations involved in the Halal market, such as what they do and how they are involved in the market. For example, "The Standards and Metrological Institute of Islamic Countries (SMIIC) is a standardization body for the Organization of Islamic Cooperation (OIC) and is responsible for developing Halal standards for OIC member countries."
- Section 2: Provide more information on the Halal standards developed by Brunei, Thailand, and Iran, such as what these standards are and how they compare to other standards.
- Section 4.1: The text mentions that there are differences among various Islamic schools of thought about the issue of Halal slaughtering, but it does not provide any information on what these differences are or how they affect the requirements for Halal slaughtering.
-Section 5: The text mentions that mutual recognition of Halal standards by accreditation bodies for import/export and permitted to function in the counterpart of respective regions is a workable solution, but it does not provide any information on how this can be achieved or what efforts are being made in this regard.
Author Response
|
|
Comments |
Compliance |
Line # |
|
1 |
The text states that Muslims are expected to increase to 2.2 billion by 2023, but it does not provide a source for this claim. |
Citation inserted for the sources. |
40 |
|
2 |
The text mentions a recent survey representing 38,000 Muslims, but it does not provide information on when the survey was conducted or who conducted it. |
Done |
44 |
|
3 |
The text lists several organizations involved in the Halal market, but it does not provide any information on what these organizations do or how they are involved in the market. |
Done |
72-82, 155-159 |
|
4 |
please indicate the similarities and differences in Halal food standards across different countries. |
Done |
291-303 |
|
5 |
The authors should provide a more in-depth analysis of the challenges faced by halal certification bodies in implementing halal standards, as well as potential solutions to these challenges |
Done |
515-535 |
|
6 |
The authors should state the impact of non-standardization on the economic growth of the halal food industry and international trade.
|
Thanks dear reviewer for your comment. Actually It is not in the scope of this article. It needs a separate research article to address the issue and we will soon complete separate article on this aspect. |
|
|
7 |
The authors should address the implications of labeling requirements in Halal food standards |
Done |
475-514 |
|
8 |
Provide more information on the organizations involved in the Halal market, such as what they do and how they are involved in the market |
Done in all places where needed, i.e. |
72-82, 155-159,172-176, 192-193 280-289 etc. |
|
9 |
Provide more information on the Halal standards developed by Brunei, Thailand, and Iran, such as what these standards are and how they compare to other standards |
More Information was provided. Comparison done in relevant sections |
255-256, 268-269, 280-289 |
|
10 |
The text mentions that there are differences among various Islamic schools of thought about the issue of Halal slaughtering, but it does not provide any information on what these differences are or how they affect the requirements for Halal slaughtering. |
Done. Included in the Heading “Halal Slaughtering” and Table 1, 2, 3 |
314-319, onward |
|
11 |
The text mentions that mutual recognition of Halal standards by accreditation bodies for import/export and permitted to function in the counterpart of respective regions is a workable solution, but it does not provide any information on how this can be achieved or what efforts are being made in this regard. |
Thanks dear reviewer your useful comment. Detailed information/data regarding the efforts for mutual recognition of Halal standards are unavailable. It is just a suggestion for future research and is not included in this article scope. |
|
Round 2
Reviewer 2 Report
Comments and Suggestions for Authors
The corrections have been made in the text, I have no more comments.